# Sero-Epidemiological Study of Varicella in the Italian General Population

**DOI:** 10.3390/vaccines11020306

**Published:** 2023-01-30

**Authors:** Giovanni Gabutti, Tiziana Grassi, Francesco Bagordo, Marta Savio, Maria Cristina Rota, Paolo Castiglia, Tatjana Baldovin, Francesco Napolitano, Alessandra Panico, Matilde Ogliastro, Claudia Maria Trombetta, Savina Ditommaso, Fabio Tramuto

**Affiliations:** 1National Coordinator of the Working Group “Vaccines and Immunization Policies”, Italian Society of Hygiene, Preventive Medicine and Public Health, 16030 Cogorno, Italy; 2Department of Biological and Environmental Sciences and Technologies, University of Salento, 73100 Lecce, Italy; 3Department of Pharmacy-Pharmaceutical Sciences, University of Bari “Aldo Moro”, 70121 Bari, Italy; 4Post-Graduate School of Hygiene and Preventive Medicine, University of Ferrara, 44121 Ferrara, Italy; 5Department of Infectious Diseases, Italian Institute of Health (ISS), 00161 Roma, Italy; 6Department of Medicine, Surgery and Pharmacy, University of Sassari, 07100 Sassari, Italy; 7Hygiene and Public Health Unit, Department of Cardiac, Thoracic, Vascular Sciences and Public Health, University of Padua, 35131 Padua, Italy; 8Department of Experimental Medicine, University of Campania “Luigi Vanvitelli”, 80138 Naples, Italy; 9Department of Health Sciences, University of Genova, 16126 Genova, Italy; 10Department of Molecular and Developmental Medicine, University of Siena, 53100 Siena, Italy; 11Department of Sciences of Public Health and Pediatrics, University of Turin, 10126 Turin, Italy; 12Department of Health Promotion, Mother and Child Care, Internal Medicine and Medical Specialties “G. D’Alessandro”, University of Palermo, 90133 Palermo, Italy

**Keywords:** varicella, sero-epidemiology, Italy, general population, immunization

## Abstract

The aim of this study was to analyze the seroprevalence of varicella in Italy and to evaluate the impact of varicella vaccination, which has been mandatory for newborns since 2017. The levels of VZV-specific IgG antibodies were determined by the ELISA method in residual serum samples obtained from subjects aged between 6 and 64 years and residing in 13 Italian regions. Overall, 3746 serum samples were collected in the years 2019 and 2020. The overall seroprevalence was 91.6% (89.9% in males and 93.3% in females; *p* = 0.0002). Seroprevalence showed an increasing trend (*p* < 0.0001) starting in the younger age groups: 6–9 years: 84.1%; 10–14 years: 88.7%; 15–19 years: 89.3%; 20–39 years: 93.1% and >40 years: 97.0%. The seroprevalence data obtained in the present study were compared with those relating to previous sero-epidemiological surveys conducted, respectively, in the years 1996–1997, 2003–2004 and 2013–2014, taking into consideration only data from regions monitored in all surveillance campaigns. The comparison highlighted for the period 2019–2020 showed significantly higher values in the age groups 6–9 (*p* < 0.001), 10–14 (*p* = 0.018) and 15–19 years (*p* = 0.035), while in adults, the trend did not change over time (ns). These results highlight the positive impact of varicella vaccination in Italy.

## 1. Introduction

Varicella (chickenpox) occurs by primary infection with varicella zoster virus (VZV) and is a widespread infectious disease that, in the pre-vaccination era, had an endemic-epidemic trend and typically affected childhood [1]. From a clinical point of view, although often considered an absolutely benign pathology, varicella can correlate with even severe complications, especially in newborns, adults and immuno-compromised subjects. Furthermore, primary VZV infection in pregnancy can result in clinically very severe forms for both the pregnant woman and the fetus/newborn [2]. The availability and extensive use of vaccination have had a significant impact on the epidemiology of varicella, directly related to the vaccination coverage rate achieved [3]. 

In the position paper published in 2014, the World Health Organization (WHO) emphasized that the VZV virus, widespread worldwide, in the absence of a vaccination program, would infect most people by young adulthood, even if the age of varicella acquisition had some geographic variation with higher values in many tropical regions. In pre-vaccinal high-income temperate countries, >90% of infections occurred before adolescence, and <5% of adults remained susceptible to VZV. In tropical areas, the acquisition of the infection occurred, for reasons not yet fully understood, at an older age with a consequent greater susceptibility among young adults. Varicella showed strong seasonality in temperate and most tropical areas, with incidence peaks during winter and spring or in drier/cooler months/periods in the tropics. Additionally, in the pre-vaccination era, epidemic peaks were recorded with a cycle of about 2–5 years [4].

The incidence in Europe was analyzed by the European Center for Disease Prevention and Control (ECDC) in 2015, highlighting how varicella surveillance systems in Europe are highly heterogeneous or lacking in several countries, resulting in a large underestimation of the real epidemiology. In the pre-vaccination era, the annual number of varicella cases was estimated to be close to each country’s birth cohort. Varicella mainly affected the younger age groups; 52–78% of incident cases occurred in children < 6 years of age, and 89–95.9% of cases occurred before adolescence. Varicella was a common childhood infection, and the annual incidence was 1580–12,124 cases/100,000 inhabitants in children aged 1–4 years and 4400–18,600/100,000 inhabitants in children 0–4 years of age, with differences in different countries [1]. 

In Italy, varicella is subject to mandatory notification, and all reported cases are registered by the Ministry of Health and the Italian Institute of Statistics (ISTAT). The overall standardized annual rate for the years 1991–2004 ranged from 164.4 to 244.2/100,000 inhabitants. By analyzing the data divided into three periods (1991–1995, 1996–2000 and 2001–2004), it clearly emerged that varicella mainly affected children (0–14 years) and that, in this age group, the incidence increased significantly (from 996/100,000 inhabitants in 1991–1995 to 1164 in 2001–2004), while it had decreased in other age groups. The comparison between subjects aged between 0 and 14 years with those > 15 years of age showed a significant increase in the percentage of cases in the younger age group (81.4% in 1991–1995, 85.2% in 1996–2000 and 88.4% in 2001–2004). Furthermore, the analysis by geographical area showed that, although the incidence was similar for the three areas (Northern, Central, and Southern Italy and Islands), there was a clear north-south gradient, with the highest incidence consistently found in Northern Italy, followed by Central and then Southern Italy [5]. 

Active and passive surveillance systems are important for evaluating the epidemiology of an infectious disease, and the national notification system adopted in Italy, being mandatory, is useful for describing the epidemiology of a specific infectious disease and for evaluating temporal trends but is certainly affected by under-reporting and under-diagnosis, as confirmed by other studies [6]. An analysis relating to the period 2001–2010 revealed an average annual number of cases equal to 88,778, with an average annual incidence equal to 150.7 cases/100,000 inhabitants. The highest incidence rate was recorded in the 0–14 age group (948.6 cases/100,000 inhabitants), which accounted for 88.8% of the total cases. The incidence rates were 41.8, 19.4 and 1.2/100,000 inhabitants in the 15–24, 25–64 and >65 age groups, respectively [7]. Notably, in this period, it was already possible to evaluate the impact of universal vaccination for varicella, which had been adopted by three regions: Sicily (in 2003), Veneto (in 2005) and Apulia (in 2006). Vaccination coverage rates increased from 40.4% in 2003 to 81.5% in 2010 in Sicily, from 68.1% in 2006 to 79.4% in 2010 in Veneto and from 43.8% in 2006 to 75.6% in 2010 in Apulia. In Sicily, the incidence of varicella decreased from 105.7 in 2003 to 9.2/100,000 inhabitants in 2010; similarly, in Veneto, the incidence decreased from 225.5 in 2007 to 55.7/100,000 inhabitants in 2010, while in Apulia, the incidence decreased from 121.7 in 2006 to 13.1/100,000 inhabitants in 2010 [7]. 

Vaccination for varicella in Italy has been included in the National Vaccine Prevention Plan (PNPV) 2017–2019, which is still in force [8]. Vaccination includes a two-dose schedule for newborns and catch-up interventions for susceptible subjects [9]. The Decree Law of 7 June 2017, n.73, *“Urgent provisions on vaccinal prevention*”, amended by the Conversion Law of 31 July 2017, n.119, made varicella vaccination mandatory for subjects aged between zero and sixteen years and for foreign minors (together with measles, rubella and mumps and the hexavalent vaccine DTaP-HBV-Hib-Polio) [10]. 

As a result of mandatory vaccination, coverage rates increased, and vaccination coverage at 24 months of age in the 2018 birth cohort measured in 2020 was 90.28% (40.56% for the 2013 cohort at 5–6 years of age). Vaccination coverage at 24 months of age in the 2019 birth cohort evaluated in 2021 increased further to 92.08% (48.40% for the 2014 cohort at 5–6 years of age) [11]. This mandatory act has effectively contributed to achieving a significant increase in coverage rates in childhood [12]. 

In this context, it is important to acquire new updates on the prevalence of subjects susceptible to varicella and to verify the impact of the vaccinal interventions adopted. To this end, a sero-epidemiological survey was conducted on a representative sample of the Italian population.

## 2. Materials and Methods

The study was designed as an in vitro, not interventional, multicenter study promoted by the Italian Institute of Health (ISS), with the aim to analyze sera collected in several Italian regions. 

Anonymous unlinked samples of residual sera from routine laboratory testing were collected from subjects between 6 and 90 years of age without any immune-depressive condition or any acute infection or who have not recently undergone blood transfusion and stored at −20°C.

Age, gender and geographical area of residence were the only demographic data collected. The number of sera to be collected was calculated, taking into account that the study was designed in order to evaluate the seroprevalence against a number of vaccine-preventable infectious diseases. In accordance with the estimates made in the course of the sero-epidemiological studies conducted within the European European Sero-Epidemiological Network (ESEN) project [13], on which samples were taken for the national seroprevalence studies conducted in 1996, 2003–2004 and 2013–2014, the sampling protocol for each regional center is summarized in Table 1. The representativeness of the sample was pursued following indications provided by the ESEN project [13]; the inclusion and exclusion criteria adopted should have allowed avoiding an overestimation of seroprevalence.

Samples were collected in the period June 2019–May 2020 from 13 regional centers (Northern Italy: Autonomous Province of Bolzano, Emilia-Romagna, Liguria, Piedmont and Veneto; Central Italy: Tuscany and Marche; Southern Italy and Islands: Basilicata, Calabria, Campania, Apulia, Sardinia and Sicily). The objectives of the study were to evaluate the prevalence of antibodies to VZV (IgG anti-VZV) in the study population by age group, gender and geographical area (Northern, Central and Southern Italy).

All collected sera were analyzed at the Laboratory of Hygiene of the Department of Biological and Environmental Sciences and Technologies, University of Salento, Lecce, Italy. 

The Enzygnost anti-VZV/IgG enzyme immunoassay (Siemens Healthcare Diagnostic Products GmbH, Germany) was used to determine the levels of VZV-specific IgG antibodies in sera. The intensity of the colorimetric reaction was determined in duplicate by the Labtech Microplate Reader LT4000 (Labtech International LTD, United Kingdom) at 450 nm using a 620 nm reference wavelength. An absorbance > 0.2 was taken as positive and indicated immunity to VZV infection; values < 0.1 were considered negative and indicated susceptibility to VZV infection. Values between 0.1 and 0.2 obtained twice on the same samples were classified as “equivocal”. The absorbance of samples tested as positive was converted into antibody concentration using an algorithm provided by the manufacturer and based on the α-method. The antibody levels were expressed as mIU/mL, based on the WHO international standard for varicella zoster immunoglobulin (50 IU). In accordance with what was declared by the manufacturer, the sensitivity and specificity of the test were equal to 99.3% and 100%, respectively.

The main statistical techniques that apply to observational studies (epidemiological statistics) were used to analyze the data collected in the study using MedCalc Software version 12.3 (MedCalc Software bvba, Ostend, Belgium).

In particular, the prevalence, expressed as a percentage of immunized subjects, and the relative 95% confidence interval (CI) were calculated in each group. The evaluation of any difference in the prevalence among age groups was carried out using the chi-square test. The same test was used to compare data stratified according to geographical area (Northern, Central and Southern Italy). In all cases, the significance level was set at 0.05.

## 3. Results

Overall, in the years 2019 and 2020, 3746 samples were collected from subjects aged between 6 and 64 years and residing in 13 Italian regions: 1439 (38.4%) came from Northern Italy (Autonomous Province of Bolzano, Piedmont, Liguria, Veneto and Emilia-Romagna), 368 (9.8%) from Central regions (Marche and Tuscany) and 1939 (51.8%) from Southern Italy and Islands (Campania, Basilicata, Apulia, Calabria, Sicily and Sardinia) (Table 2). The small sample size in Central Italy is related to the fact that only 2 regions were included in the study.

Subjects were stratified into 5 age groups: 376 (10.0%) were aged ≤ 9 years, 571 (15.2%) were aged between 10 and 14 years, 556 (14.8%) were aged between 15 and 19 years, 1698 (45.3%) were aged between 20 and 39 years, and 545 (14.5%) were aged between 40 and 64 years.

Of the samples tested, 3362 were positive, and 308 were negative. The remaining 76 sera, confirmed as equivocal, were excluded from the analysis. Thus, the overall seroprevalence was 91.6% (95% CI = 90.7–92.5%), with a significant difference between males (89.9%; 95% CI = 88.4–91.2%) and females (93.3%; 95% CI = 92.1–94.4%)) (*p* < 0.001).

Seroprevalence showed a significantly increasing trend (*p* < 0.001) starting from the lowest age groups: 6–9 years: 84.1%; 10–14 years: 88.7%; 15–19 years: 89.3%; 20–39 years: 93.1% and >40 years: 97.0% (Figure 1). As highlighted by the 95% CI analysis (Table 3), the differences were detected between the first three age groups (6–9 years, 10–14 years and 15–19 years) and the older age groups (20–39 years and >40 years).

The distribution by geographical area shows an overall seroprevalence higher in Northern Italy (93%) and lower in Central (91.4%) and Southern Italy (90.6%). These differences were not significant (*p* = 0.583) (Figure 2).

However, significant differences were observed when analyzing seroprevalence in the different age groups also stratified by geographical area (Table 3). In Northern Italy, a seroprevalence of 77.1% was recorded in the 6–9 years age group, significantly lower (*p* = 0.0106) than in other geographical areas. In the following age groups, seroprevalence in Northern Italy gradually increased up to 97.8% in the 40–64 age group. On the contrary, in Southern Italy, the seroprevalence declined from 88.8% in the 6–9 years-old class to 84.7% in the 10–14 years-old class and then increased in the following age groups. In Central Italy, the trend did not show significant differences between different age groups.

It is interesting to compare the seroprevalence recorded in areas where varicella vaccination was started at different times (Table 4). Among the regions participating in the study, 7 (Basilicata, Calabria, Apulia, Sardinia, Sicily, Tuscany and Veneto) had progressively introduced (from 2003, Sicily, to 2011, Sardinia) universal vaccination and are therefore considered “pilot regions” (in addition to these, Friuli Venezia Giulia, a region not participating in the study, also introduced universal varicella vaccination in 2013). The other regions (Autonomous Province of Bolzano, Campania, Emilia-Romagna, Liguria, Marche and Piedmont) have adhered to the 2017–2019 National Vaccine Prevention Plan, which provided for the introduction of varicella vaccination for all newborns from the birth cohort of 2016.

The results of the current sero-epidemiological survey (conducted between 2019 and 2020) show significant differences in relation to the distribution of seroprevalence by age group between the two groups of regions. In particular, in the pilot regions in the 6–9 years age group, the prevalence was equal to 89.3% versus 73.9% recorded in the regions that implemented universal vaccination starting from those born in 2016 (subjects not included in the 6–9 age group). 

This also justifies the different distribution by age group found among the various Italian geographical areas, considering that 5 out of 6 regions of Southern Italy and 1 out of 2 regions of Central Italy belong to the pilot regions, while only 1 out of 5 regions of Northern Italy has started universal varicella vaccination early.

Seroprevalence data obtained in the present study were compared with those relating to previous sero-epidemiological surveys conducted, respectively, in the years 1996–1997 [14], 2003–2004 [5], and 2013–2014 [15], taking into consideration only the subjects residing in the regions monitored in all surveillance campaigns (Autonomous Province of Bolzano, Calabria, Emilia-Romagna, Liguria, Piedmont, Apulia, Sardinia, Sicily, Tuscany and Veneto) (Table 5). The statistical analysis (chi-square test) highlighted a significant difference, particularly evident in the 6–9 years age group (*p* < 0.001), more contained in the 10–14 (*p* = 0.018) and 15–19 years age group (*p* = 0.035), while in adults, the trend is quite similar (*p* > 0.05).

## 4. Discussion

Sero-epidemiology, understood as the collection of data on the prevalence of antibodies in the serum in a population, represents a fundamental tool that allows not only an evaluation of the interventions implemented in the past but also of the contingent risk of infectious diseases. This type of surveillance is particularly useful for vaccine-preventable diseases for which, by measuring the results achieved, alternative immunization strategies can be employed or, conversely, those already in place can be implemented. Through serological investigations, it is possible to identify groups of individuals susceptible to a certain infectious disease that could be involved in epidemic outbreaks, allowing targeted preventive interventions [16].

Several seroprevalence studies have been conducted in Italy with the aim of both confirming data deriving from other active and passive surveillance systems and verifying the impact of the vaccinal interventions adopted over the years. In the survey performed in 2013–2014, it was highlighted that seroprevalence had a typical trend with a decrease in the first year of life in subjects previously passively protected by the mother and a subsequent increase in the following age groups (0–11 months: 47.2%; 1 year: 25.8%; 2–4 years: 47.0%; 5–9 years: 75.3%; 10–14 years: 89.4%; 15–19 years: 89.8%; 20–39 years: 93.2% and >40 years: 98.4%). The overall seroprevalence was 84.0%, without any statistically significant difference by gender and with a significant difference between the three Italian geographical areas (North, 84.9%; Central, 87.3%; and South, 80.6%). The overall seroprevalence detected in the study conducted in the years 2013–2014 (84%) was higher than that observed in two previous studies conducted with the same methodology in 1996–1997 (73.2%) and in 2003–2004 (77.8%). Furthermore, compared to the data obtained in 1996–1997 and in 2003–2004, the seropositivity rate found in 2013–2014 was significantly increased in the 1 and 2–4 age groups, highlighting the crucial role played by the immunization interventions already implemented in 8 regions which, in terms of residents, represented 37.2% of the Italian population. In the 5–19 years age group, seroprevalence was higher in the remaining regions, probably due to the high circulation of VZV in the absence of a universal immunization strategy [15]. The data of the present work, even if based on sera collected starting from 6 years of age, show an overall seroprevalence equal to 91.6%, higher than the one previously found. It is relevant the significant increase in the seroprevalence data in the 6–9 years age group as it includes subjects targeted by the vaccinal interventions adopted over the last few years. These data are in line with what has been observed in different areas of the world; the introduction of vaccination and the achievement of high coverage rates correlates with a decrease in morbidity, hospitalizations and mortality and with changes in seroprevalence data [17,18,19,20,21,22,23]. Unfortunately, the COVID-19 pandemic has caused a decline in coverage rates worldwide. In Italy, a decrease in hexavalent, measles-mumps-rubella-varicella, meningococcal and pneumococcal vaccines has been registered in the 0–2 years age group, as well as a decline in human papillomavirus (HPV) coverage rates in adolescents. A great effort should be made in order to plan the recovery of vaccinations not performed during the different waves of COVID-19 and to strengthen immunization programs [24].

This study has some strengths as well as some limitations. A strength is that the adopted methodology is exactly the same one used during previous studies performed in Italy as far as sampling and ELISA tests are taken into account. This point guarantees the comparability of data collected in different periods. A possible limitation is related to the fact that the study is based on residual serum samples (convenience samples), which may not be representative of the general population. However, the obtained data nonetheless provide an indication of the seroprevalence rate in the various age groups, as expected from studies on large samples. Finally, the unavailability of data on previous vaccination prevents the distinction of positivity deriving from natural infection or from immunization.

In conclusion, the implementation of extensive varicella vaccination worldwide has had a significant impact on public health, with up to 80% reductions in incidence, hospitalizations, and varicella-related complications [3]. However, some barriers remain to be overcome. There is a need to improve epidemiological surveillance, to make available and to disseminate knowledge on the impact of already-implemented vaccination programs, and to better communicate the risks and benefits of varicella vaccination to policymakers, health professionals and the general public in order to achieve the preventative goals and to ensure that chickenpox is no longer a public health problem. In this context, sero-epidemiological studies are a useful tool not only to evaluate the distribution of susceptible subjects in the population but also to evaluate the impact of extensive vaccination.

## Figures and Tables

**Figure 1 vaccines-11-00306-f001:**
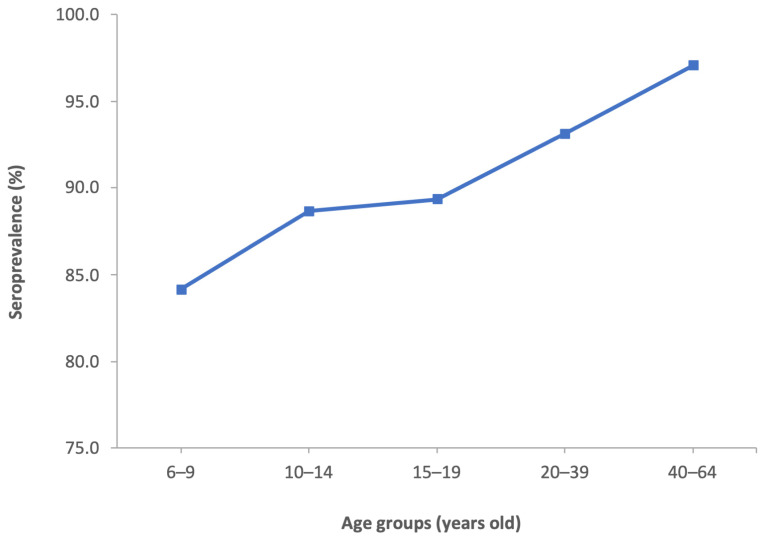
Seroprevalence of anti-VZV antibodies by age group in the Italian population.

**Figure 2 vaccines-11-00306-f002:**
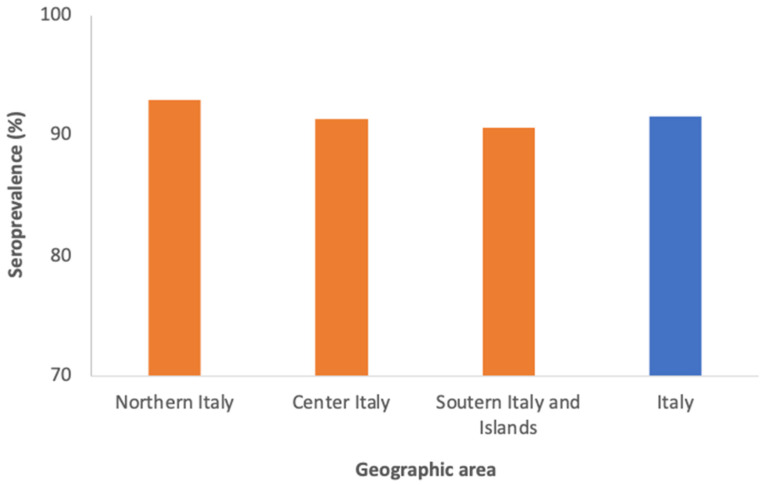
Seroprevalence of anti-varicella antibodies in the Italian geographical areas.

**Table 1 vaccines-11-00306-t001:** Sample size for each center.

Age (Years)	Sample Size	Total
**6–20**	Each year, equally distributed between males and females	**165**
**21–40**	Quinquennial age classes, equally distributed between males and females	**140**
**41–50**	10 (5 males, 5 females)	**10**
**51–64**	16 (8 males, 8 females)	**16**
**65–74**	16 (8 males, 8 females)	**16**
**>75**	16 (8 males, 8 females)	**16**
**Total**		**363**

**Table 2 vaccines-11-00306-t002:** Distribution of samples stratified by geographical area and age group.

	Age Group (Years Old)					Total
	6–9	10–14	15–19	20–39	40–64	
**Northern Italy**	146	260	235	614	184	1439
**Central Italy**	23	45	58	179	63	368
**Southern Italy and Islands**	207	266	263	905	298	1939
**Total**	376	571	556	1698	545	3746

**Table 3 vaccines-11-00306-t003:** Seroprevalence (%) and relative 95% CI by age group in the Italian geographical areas.

Age Groups (Years)	Northern	Central	Southern and Islands	Italy	*p*-Value ***
6–9	77.1 (70.2–83.9)	90.5 (77.0–100.0)	88.8 (84.3–93.3)	**84.1** **(80.3–87.9)**	0.0106
10–14	91.8 (88.4–95.1)	93.0 (85.4–100.0)	84.7 (80.3–89.2)	**88.7** **(86.0–91.3)**	0.0292
15–19	93.0 (89.6–96.3)	91.1 (83.6–98.5)	85.8 (81.6–90.5)	**89.3** **(86.7–91.9)**	0.0356
20–39	95.7 (94.1–97.3)	89.4 (84.9–93.9)	92.1 (90.3–93.8)	**93.1** **(91.9–94.3)**	0.0025
40–64	97.8 (95.7–99.9)	96.8 (92.4–100.0)	96.6 (94.6–98.7)	**97.0** **(95.6–98.5)**	0.7577
**TOTAL**	**93.0** **(91.6–94.3)**	**91.4** **(88.5–94.3)**	**90.6** **(89.3–91.9)**	**91.6** **(90.7–92.5)**	0.5832
*p*-value *	<0.0001	0.4961	<0.0001	<0.0001	

* *p*-value calculated by chi-square test.

**Table 4 vaccines-11-00306-t004:** Seroprevalence (%) and relative 95% CI by age group recorded in the pilot regions and in the regions that started varicella vaccination from the cohort of those born in 2016.

Age Groups (years)	Pilot Regions ^1^	Remaining regions ^2^	Total	*p*-Value *
6–9	89.3 (85.4–93.3)	73.9 (66.1–81.8)	84.1 (80.3–87.9)	<0.001
10–14	86.9 (83.3–90.6)	91.3 (87.5–95.0)	88.7 (86.0–91.3)	0.116
15–19	86.5 (82.8–90.2)	93.3 (90.2–96.5)	89.3 (86.7–91.9)	0.011
20–39	91.8 (90.3–93.3)	94.7 (93.3–96.1)	93.1 (91.9–94.3)	0.021
40–64	96.0 (93.7–98.3)	98.1 (96.5–98.8)	97.0 (95.6–98.5)	0.148

^1^ Basilicata, Calabria, Apulia, Sardinia, Sicily, Tuscany, and Veneto. ^2^ Autonomous Province of Bolzano, Campania, Emilia-Romagna, Liguria, Marche, and Piedmont. * Calculated by chi-square test.

**Table 5 vaccines-11-00306-t005:** Comparison of seroprevalence recorded in different studies conducted in Italy.

Age Groups (Years)	Seroprevalence (%)				*p*-Value *
	1996/97 Ref. [13]	2003/04 Ref. [14]	2012/13 Ref. [5]	2019/20	
6–9	57.9	61.6	63.3	88.4	<0.001
10–14	80.2	83.3	84.8	88.8	0.018
15–19	81.4	85.7	86.0	89.2	0.035
20–39	91.3	90.6	90.9	93.3	0.175
40–64	99.2	97.6	98.3	96.0	0.161
TOTAL	81.3	84.6	85.4	91.9	<0.001

* Calculated by chi-square test

## Data Availability

Not applicable.

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
