# Peer review of "Sero-Epidemiological Study of Varicella in the Italian General Population"

_vaccines, 2023, doi:10.3390/vaccines11020306_

Round 1

Reviewer 1 Report

Giovanni et al., analyze the seroprevalence of varicella in Italy and to evaluate the impact of varicella vaccination. Authors collected 3746 samples (with 76 equivocal samples) from aged between 6 and 64 years and residing in 13 regions. Furthermore, this data was compared with previous sero-epidemiological surveys. It is an interesting study, and some concerns need to pay attention.

1) The statistical methods in the materials and methods section should be introduced in more detail, including the selection of confidence intervals and other information needs to be indicated. In addition, do you use other statistical algorithms except chi square test?

2) Table 2: The underline of northern Italy is incorrect.

3) Table 2: The samples size in Center Italy was significant smaller than Northern Italy and Southern Italy and Islands but with similar age distribution ratio. Authors should mention the reasons for the small sample size in the central region of Italy, such as: including two regions.

4) Fig 1 and Fig 2: the statistical methods should be mentioned clearly in the results. If the chi square test is used for comparison between two groups, it is necessary to specify which two groups are compared.

5) Table 3-Table 5: Also the statistical methods used in these results should be mentioned more clearly.

Author Response

Giovanni et al., analyze the seroprevalence of varicella in Italy and to evaluate the impact of varicella vaccination. Authors collected 3746 samples (with 76 equivocal samples) from aged between 6 and 64 years and residing in 13 regions. Furthermore, this data was compared with previous sero-epidemiological surveys. It is an interesting study, and some concerns need to pay attention.

1) The statistical methods in the materials and methods section should be introduced in more detail, including the selection of confidence intervals and other information needs to be indicated. In addition, do you use other statistical algorithms except chi square test?
Thank you. More details have been added in the Materials and methods section including the software used to perform the statistical analysis. Information related to confidence interval calculation have also been added (page 4, lines 162-166)

2) Table 2: The underline of northern Italy is incorrect.      
Thank you. The underline has been deleted.

3) Table 2: The samples size in Center Italy was significant smaller than Northern Italy and Southern Italy and Islands but with similar age distribution ratio. Authors should mention the reasons for the small sample size in the central region of Italy, such as: including two regions.
Thank you. A sentence has been added in page 4, lines 177-178 in order to clarify this point.

4) Fig 1 and Fig 2: the statistical methods should be mentioned clearly in the results. If the chi square test is used for comparison between two groups, it is necessary to specify which two groups are compared.    
Thank you. A comment has been added in page 5, lines 189-191.

5) Table 3-Table 5: Also the statistical methods used in these results should be mentioned more clearly.        
Thank you. More details regarding the statistical analysis have been added in the text as well as in the table captions.

Reviewer 2 Report

The authors of the paper I was kindly asked to review conducted a sero-epidemiological study to analyse the seroprevalence of varicella in the Italian general population and to evaluate the impact of varicella vaccination, which is mandatory for newborns starting since 2017.

The paper is well structured; the methodology is clearly stated.

Some minor revisions are reported below:

-       In the introduction, concerning effect of mandatory vaccination, reference could be made to further studies that have addressed the topic (i.e., please consider citing Sabbatucci M, Odone A, Signorelli C, Siddu A, Maraglino F, Rezza G. Improved Temporal Trends of Vaccination Coverage Rates in Childhood after the Mandatory Vaccination Act, Italy 2014-2019. J Clin Med. 2021 Jun 8;10(12):2540. doi: 10.3390/jcm10122540. PMID: 34201199; PMCID: PMC8230222.)

-       Regarding methods, it might be useful to describe any efforts to address potential sources of bias.

-       In the discussion, it could be useful to comment the latest trends of vaccination coverage after COVID-19 pandemic.

-       I believe authors could also describe how missing data were addressed.

-       In table 5, it could be helpful, if possible, to include data on overall seroprevalence.

-       In table 5, please pay attention to commas instead of full stops.

-       In table 5, please consider the difference in the data in the column referring to 2019/2020 compared to the previous two tables.

Author Response

The authors of the paper I was kindly asked to review conducted a sero-epidemiological study to analyse the seroprevalence of varicella in the Italian general population and to evaluate the impact of varicella vaccination, which is mandatory for newborns starting since 2017.

The paper is well structured; the methodology is clearly stated.  
Thank you for this good evaluation of our manuscript.

Some minor revisions are reported below:

-       In the introduction, concerning effect of mandatory vaccination, reference could be made to further studies that have addressed the topic (i.e., please consider citing Sabbatucci M, Odone A, Signorelli C, Siddu A, Maraglino F, Rezza G. Improved Temporal Trends of Vaccination Coverage Rates in Childhood after the Mandatory Vaccination Act, Italy 2014-2019. J Clin Med. 2021 Jun 8;10(12):2540. doi: 10.3390/jcm10122540. PMID: 34201199; PMCID: PMC8230222.)     
Thank you. The suggested reference has been added (Ref 12) as well as a brief sentence in page 3, lines 113-114. The reference numbering has been changed accordingly.

-       Regarding methods, it might be useful to describe any efforts to address potential sources of bias.         
Thank you. A sentence on this point has been included in page 3, lines 134-137.

-       In the discussion, it could be useful to comment the latest trends of vaccination coverage after COVID-19 pandemic.

Thank you. A sentence has been added in page 8, lines 285-290. A reference has been added as well (Ref 25).

-       I believe authors could also describe how missing data were addressed.

Thank you. We are not sure to have properly understood this point. Limitations of the study have already been included in the Discussion section (page 8-9, lines 295-300).

-       In table 5, it could be helpful, if possible, to include data on overall seroprevalence.

Thank you. As requested, data on overall seroprevalence have been added.

-       In table 5, please pay attention to commas instead of full stops.

Thank you. Commas have been included.

-       In table 5, please consider the difference in the data in the column referring to 2019/2020 compared to the previous two tables.

Thank you. Please take note that data included in table 5 refer only to the regions that have been included also in the previous surveillances (Autonomous Province of Bolzano, Calabria, Emilia-Romagna, Liguria, Piedmont, Apulia, Sardinia, Sicily, Tuscany and Veneto). For this reason, it seems not completely correct to compare data included in table 5 with all data included in other studies/tables, as previous studies included regions that have not been evaluated in the present research. For the above-mentioned reasons, we would like to not change anything in table 5.

Reviewer 3 Report

                 Gabutti G, et al. conducted a seroepidemiological study after national varicella vaccination in Italy. Compared to previous studies, seroprevalence in adults did not change, but seroprevalenc increased in those aged 19 and younger.

General comments:

              As the authors showed in the manuscript, it is important to know the seroepidemiology of vaccine preventable diseases. In order to indicate the effect of varicella immunization more in this study, it would be helpful to describe the age of national immunization and catch up immunization in Italy. And based on the information, can you assign the group as generation taking national immunization, taking catch up immunization and the other group? This would make it easier to discuss whether the increase in seroprevalence is due to natural varicella infection, reactivation, or vaccine-induced. In the present data, it is unclear whether the increase of seroprevalence is due to natural or vaccine-induced infection.

              Moreover, please mention about past history of varicella in subjects of the present study. This would allow discussion  whether it is due to positivity of antibody due to vaccines or natural infection.

Minor comment

Page 1, Line 40: “Varicella (chickenpox) occurs by the primary infection,,,” may be better as description.

Page 2, Line 76: “it had decreased in other age group” may be better as description.

Page 4, Line 144: What dose “Enzygnost” mean?

Methods: How did you transfer from OD titer to IU?

Page 7: Please add references about these previous seroepidemiology in Italy.

Page 7, Line 254-257: Unclear what this sentence means. Can you explain the sentence in another way?

Author Response

Gabutti G, et al. conducted a seroepidemiological study after national varicella vaccination in Italy. Compared to previous studies, seroprevalence in adults did not change, but seroprevalence increased in those aged 19 and younger.

General comments:
As the authors showed in the manuscript, it is important to know the seroepidemiology of vaccine preventable diseases. In order to indicate the effect of varicella immunization more in this study, it would be helpful to describe the age of national immunization and catch up immunization in Italy. And based on the information, can you assign the group as generation taking national immunization, taking catch up immunization and the other group? This would make it easier to discuss whether the increase in seroprevalence is due to natural varicella infection, reactivation, or vaccine-induced. In the present data, it is unclear whether the increase of seroprevalence is due to natural or vaccine-induced infection.

Moreover, please mention about past history of varicella in subjects of the present study. This would allow discussion whether it is due to positivity of antibody due to vaccines or natural infection.              
Thank you. The points raised by the Reviewer are absolutely correct and relevant. However, the description of Italian immunization strategy against varicella has already been included in the Introduction section (pages 2-3, lines 70-118). Besides, in the Discussion section we have already highlighted that a limitation of this study is related to the fact that “the unavailability of data on previous vaccination prevents the distinction of positivity deriving from natural infection or from immunization.” (page 8-9, lines 298-299)

For these reasons, we would like to not add nor change anything in both Introduction and Discussion sections.

Minor comment

Page 1, Line 40: “Varicella (chickenpox) occurs by the primary infection,,,” may be better as description.      
Thank you. We have changed this part as suggested.

Page 2, Line 76: “it had decreased in other age group” may be better as description.
Thank you. We have changed this part as suggested.

Page 4, Line 144: What dose “Enzygnost” mean?      
Thank you. Enzygnost is commercial name of the ELISA test used to determine the levels of VZV-specific IgG antibodies in sera.

Methods: How did you transfer from OD titer to IU?         
Thank you. In the text we have specified more clearly the way we transfer from OD titer to IU: “The absorbance of samples tested as positive was converted into antibody concentration using an algorithm provided by the manufacturer and based on the α-method.” (page 4, lines 155-157)

Page 7: Please add references about these previous seroepidemiology in Italy.
Thank you. References have already been included (page 7, lines 238-243): “Seroprevalence data obtained in the present study were compared with those relating to previous sero-epidemiological surveys conducted respectively in the years 1996-97 [14], 2003-04 [15], and 2013-14 [16] taking into consideration only the subjects residing in the regions monitored in all surveillance campaigns (Autonomous Province of Bolzano, Calabria, Emilia-Romagna, Liguria, Piedmont, Apulia, Sardinia, Sicily, Tuscany and Veneto) (Table 5).”

Page 7, Line 254-257: Unclear what this sentence means. Can you explain the sentence in another way?
Thank you. Please take note that data included in table 5 refer only to the regions that have been included also in the previous studies (Autonomous Province of Bolzano, Calabria, Emilia-Romagna, Liguria, Piedmont, Apulia, Sardinia, Sicily, Tuscany and Veneto). For this reason, it seems correct to compare data belonging to regions that have been studied in all our sero-epidemiological studies. For the above-mentioned reasons, we would like to not change this part.

Round 2

Reviewer 3 Report

This revised manuscript has been properly responded to and revised in accordance with the reviewers' comments.